Morphology of an Early Oligocene beaver Propalaeocastor irtyshensis and the status of the genus Propalaeocastor

Li Lüzhou 1 2
Li Qiang liqiang@ivpp.ac.cn 1 2 3
Lu Xiaoyu 1 2
Ni Xijun nixijun@ivpp.ac.cn 1 2 3
1 Key Laboratory of Vertebrate Evolution and Human Origins of Chinese Academy of Sciences, Institute of Vertebrate Paleontology and Paleoanthropology, Chinese Academy of Sciences , Beijing , China
2 University of Chinese Academy of Sciences , Beijing , China
3 CAS Center for Excellence in Tibetan Plateau Earth Sciences , Beijing , China
Cox Philip
Electronic publication date: 2017 May 16
Publication date: 2017
Volume: 5
Electronic Location ID: e3311
Received 2016 Apr 23; Accepted 2017 Apr 13
Copyright: ©2017 Li et al.
Copyright year: 2017
Copyright holder: Li et al.
License: This is an open access article distributed under the terms of the Creative Commons Attribution License, which permits unrestricted use, distribution, reproduction and adaptation in any medium and for any purpose provided that it is properly attributed. For attribution, the original author(s), title, publication source (PeerJ) and either DOI or URL of the article must be cited.
License URL: https://creativecommons.org/licenses/by/4.0/

Keywords: Xinjiang, Beavers, Propalaeocastor, Early Oligocene, Irtysh River Formation

Funding: trategic Priority Research Program of Chinese Academy of Sciences CAS, XDB03020000 National Basic Research Program of China 2012CB821904 CAS 100-talent Program National Natural Science Foundation of China 41472025 to XN 41472002 to QL This project was supported by the Strategic Priority Research Program of Chinese Academy of Sciences (CAS, XDB03020000), the National Basic Research Program of China (2012CB821904), the CAS 100-talent Program, the National Natural Science Foundation of China (41472025 to XN, and 41472002 to QL). The funders had no role in study design, data collection and analysis, decision to publish, or preparation of the manuscript.

==============================
The Early to Late Oligocene Propalaeocastor is the earliest known beaver genus from Eurasia. Although many species of this genus have been described, these species are defined based on very fragmentary specimens. Propalaeocastor irtyshensis from the Early Oligocene Irtysh River Formation in northwestern Xinjiang, China is one of the earliest-known members of Propalaeocastor. This species is defined on a single maxillary fragment. We revise the diagnosis of P. irtyshensis and the genus Propalaeocastor, based on newly discovered specimens from the Irtysh River Formation. The dental morphology of P. irtyshensis is very similar to other early castorids. The caudal palatine foramen of P. irtyshensis is situated in the maxillary-palatine suture. This is a feature generally accept as diagnostic character for the castorids. On the other hand, P. irtyshensis has two upper premolars, a rudimentarily developed sciuromorph-like zygomatic plate, and a relatively large protrogomorph-like infraorbital foramen. Some previous researchers suggested that Propalaeocastor is a junior synonym of Steneofiber, while other took it as a valid genus. Our morphological comparison and phylogenetic analysis suggest that Propalaeocastor differs from Steneofiber and is a valid genus. We also suggest that Agnotocastor aubekerovi, A. coloradensis, A. galushai, A. readingi, Oligotheriomys primus, and “Steneofiber aff. dehmi” should be referred to Propalaeocastor. Propalaeocastor is the earliest and most basal beaver. The origin place of Propalaeocastor and castorids is uncertain. The Early Oligocene radiation of castorids probably is propelled by the global climate change during the Eocene-Oligocene transition.

Introduction

Extant and fossil beavers are medium to large body-sized semi-aquatic, terrestrial or burrowing rodents (Rybczynski, 2007; Flynn & Jacobs, 2008). Extant beavers include one genus and two species (Castor fiber and C. canadensis). Fossil beavers are much more diverse, including at least twenty seven genera and more than one hundred species (McKenna & Bell, 1997; Korth & Samuels, 2015; Mörs, Tomida & Kalthoff, 2016; https://www.paleobiodb.org/). It is generally accepted that all beavers represent a monophyletic family: Castoridae (McKenna & Bell, 1997; Helgen, 2005; Rybczynski, 2007). Castoridae is closely related to the extinct family Eutypomyidae, and the two families are usually referred to the superfamily Castoroidea (Simpson, 1945; Wood, 1955; Wood, 1965; Hugueney, 1999; Flynn & Jacobs, 2008). Within crown rodents, phylogenetic analyses based on molecular data and/or morphological data usually support the sister-group relationship between the castorids and the geomyoids (a superfamily of rodents that contains the pocket gophers, the kangaroo rats and mice, e.g., Douady et al., 2000; Adkins et al., 2001; Adkins, Walton & Honeycutt, 2003; Murphy et al., 2001; Huchon et al., 2002; Montgelard et al., 2002; Fabre et al., 2012).

The earliest-known castorid fossil, “Agnotocastor” galushai, was discovered from the South Fork of Lone Tree Gulch of Wyoming (Emry, 1972). The age of the locality is middle to late Chadronian of North American Land-Mammalian Ages (NALMA) within a precision 206Pb/238U zircon dates from 35.805 ± 0.076 Ma to 34.398 ± 0.022 Ma (Emry & Korth, 2012; Sahy et al., 2015). The dental and cranial morphology of Agnotocastor shares many similarities with the eutypomsid Eutypomys (Wilson, 1949a; Wood, 1965; Wahlert, 1977; Xu, 1995; Xu, 1996; Flynn & Jacobs, 2008). The earliest-known beavers outside of the North America belong to the genus Propalaeocastor (Borisoglebskaya, 1967; Misonne, 1957; Borisoglebskaya, 1967; Lytschev, 1970; Kretzoi, 1974; Bendukidze, 1993; Lytschev & Shevyreva, 1994; Wu et al., 2004; Bendukidze, De Bruijn & Van Den Hoek Ostende, 2009).

The validity of Propalaeocastor is debatable. The type species, P. kazakhstanicus, is from the Early Oligocene of Kyzylkak, Dzhezkazgan, Kazakhstan (Borisoglebskaya, 1967). Lytschev & Shevyreva (1994), Lopatin (2003) and Lopatin (2004) considered Propalaeocastor as a junior synonym of Steneofiber Geoffroy Saint-Hilaire, 1833. Some other researchers did not agree and suggested that Propalaeocastor is different from Steneofiber and is a valid genus (McKenna & Bell, 1997; Korth, 2001; Wu et al., 2004). Kretzoi (1974) referred “Steneofiber” butselensis (Misonne, 1957) to a new genus “Asteneofiber”. However, the validity of Asteneofiber was not widely recognized. Some researchers considered “Asteneofiber” as the junior synonym of Steneofiber (McKenna & Bell, 1997), while Wu et al. (2004) regarded “Asteneofiber” as a junior synonym of Propalaeocastor.

There are quite a few species attributed to Propalaeocastor, but the species attribution of this genus is ambiguous, because all of the species are represented by isolated teeth and/or jaw fragments. Besides the type species Propalaeocastor kazakhstanicus, Borisoglebskaya (1967) also named P. habilis in the same paper. In their study of beaver remains from Maylibay of Zaissan (or Zaysan) Basin, Kazakhstan, Lytschev & Shevyreva (1994) synonymized P. habilis with P. kazakhstanicus and reported another three species: P. shevyrevae, P. aff. shevyrevae and P. zaissanensis. Wu et al. (2004) recognized P. butselensis, P. shevyrevae, P. sp. aff. P. shevyrevae, P. zaissanensis, P. kazakhstanicus, and named the species P. irtyshensis. Lopatin (2003) suggested that “Capacikala sajakensis” is the junior synonym of “Steneofiber” kumbulakensis. Bendukidze, De Bruijn & Van Den Hoek Ostende (2009) synonymized “Capacikala sajakensis” to “Capatanca” schokensis, and transferred “Capatanca” schokensis Bendukidze, 1993 and “Steneofiber” kumbulakensis Lytschev, 1970 to Propalaeocastor.

Because of the impoverishment of specimens and ambiguous generic diagnosis, the systematic position of Propalaeocastor is also in doubt. It has been assigned to the tribe Anchitheriomyini by Korth (2001), the subfamily Anchitheriomyinae by Korth (2004) and tribe Minocastorini by Mörs, Tomida & Kalthoff (2016). The handful of dental specimens of Propalaeocastor exhibit a pattern resembling both Agnotocastor and Eutypomys. For instance, one of the Propalaeocastor species (P. kumbulakensis Lytschev, 1970) was even considered a member of Eutypomys (Xu, 1996).

To clarify the validity and species attribution of Propalaeocastor, we report a few newly discovered specimens of P. irtyshensis from the Early Oligocene Irtysh River Formation in Xinjiang, China. These specimens make P. irtyshensis the best-known species of Propalaeocastor. We examine the dental features of most of the castorid genera, and develop a data matrix for phylogenetic analysis. Based on the newly collected specimens and the results of our phylogenetic analysis on castorids, we are able to emend the generic diagnosis of Propalaeocastor and clarify the phylogenetic relationships among Propalaeocastor, Agnotocastor, Eutypomys and other early beavers.

Geologic Setting

Cenozoic sediments are widely exposed in the drainage area of the Irtysh (=Ertix) River in Burqin-Jeminay region in northwestern Xinjiang of China (Figs. 1A and 1B). Propalaeocastor irtyshensis was discovered from the lower portion of the Early Oligocene Irtysh River Formation at the XJ200203 locality in the Burqin-Jeminay region (Fig. 1B) (Wu et al., 2004; Stidham & Ni, 2014). Only upper dentition was previously known. The new specimens of P. irtyshensis reported here were discovered from a new fossiliferous locality of the lower Irtysh River Formation about 50 km southwest to the XJ200203 locality. The Irtysh River Formation is a set of fluviolacustrine mudstone, siltstone, sandstone and thick conglomerate. The fossiliferous layer of the Irtysh River Formation is dated as 32.0 Ma (Sun et al., 2014). The same fossiliferous layer at the XJ200203 locality can be traced to the new locality despite the long distance between the two localities. This fossiliferous layer at the new fossil locality is an approximately 5-m thick bed of grey greenish and light brown-reddish mudstone with rich calcareous nodules (Fig. 1C). The new P. irtyshensis remains include a fragmentary maxilla, several incomplete jaws and isolated cheek teeth. The small mammals associated with these new beaver fossils include Cricetops dormitor, Parasminthus tangingoli, Cyclomylus lohensis, and Prosciurus sp. These small mammals are also present at the XJ200203 locality (Ni et al., 2007; Sun et al., 2014).

Figure 1 Jeminay and Burqin Propalaeocastor irtyshensis fossil localities in the Irtysh River drainage area in northwestern Xinjiang, China (modified from Stidham & Ni, 2014).

(A) Map showing the location of the P. irtyshensis localities in the Irtysh River region within China adjacent to several other countries. (B) Detailed map showing the border region between Xinjiang and Kazakhstan and the localities of P. irtyshensis. (C) Panoramic view of the fossiliferous profile of the Irtysh River Formation that produced the additional material of P. irtyshensis.

Materials, Methods and Abbreviations

The new materials include a broken maxilla preserving P4-M1, two isolated upper cheek teeth and three mandibular fragments. The holotype of Propalaeocastor irtyshensis (IVPP V 13690) is re-described. All fossils are housed at the Institute of Vertebrate Paleontology and Paleoanthropology, Chinese Academy of Sciences, Beijing. The specimens were CT-scanned using the 225 kV Micro-CT at the Key Laboratory of Vertebrate Evolution and Human Origins, Chinese Academy of Sciences. Segmentations and 3D virtual reconstructions were made following the standard procedure introduced by Ni, Flynn & Wyss (2012). Specimens were measured using an Olympus SZX7 microscope and mandibles by vernier caliper both with a precision of 0.01 mm. The length is defined as the mesiodistal chord. The width is defined along the chord perpendicular to the length. For incisors, the same standard is used to define the length and width.

Figure 2 The upper dental structure of the Castoridae after the example of the moderately worn fourth premolars of Propalaeocastor irtyshensis.

(A) Propalaeocastor; (B) Steneofiber; (C) Castor; (D) Dipoides. From left to right: lingual view, occlusal view, buccal view. Enamel = white, dentine = dark grey, cement = light grey, -fossette = -flexus = -stria. Modified from Stirton (1935), Hugueney (1975), Hugueney (1999), Lopatin (2003) and Wu et al. (2004).

Figure 3 The lower dental structure of the Castoridae after the example of the moderately worn fourth premolars of Propalaeocastor irtyshensis.

(A) Propalaeocastor; (B) Steneofiber; (C) Castor; (D) Dipoides. From left to right: lingual view, occlusal view, buccal view. Enamel = white, dentine = dark grey, cement = light grey, -fossettid = -flexid = -striid. Modified from Stirton (1935), Hugueney (1975), Hugueney (1999), Lopatin (2003) and Wu et al. (2004).

The dental terminology (Figs. 2 and 3) is modified from Stirton (1935), Hugueney (1975), Hugueney (1999), Lopatin (2003) and Wu et al. (2004). We use “-loph” and “-lophid” for the major ridges or crests, and “-lophule” and “-lophulid” for the thin, short spur-like ridges that are developed from the lophs and lophids. The major change is that we abandon the use of terms “mesoloph” and “mesolophid” in castorids. The mesoloph and mesolophid are usually defined as “crest from mesocone(id) toward the lingual or buccal side of the tooth” (Wood & Wilson, 1936). The mesocone and mesoconid are distinctly present in Eutypomys, and the mesoloph and mesolophid are clearly derived from the mesocone and mesoconid, respectively. In beavers, however, the mesocone and mesoconid are absent. The so-called “mesoloph(id)” is derived from the posterior arm of the protocone(id). Here we treat the so-called “mesoloph” and “mesolophid” as protoloph II and metalophid II, respectively. The dental cusp-ridge connections of the Eutypomys, Agnotocastor, Propaleocastor, and other early beavers are very complicated, i.e., their ridges are normally irregular and wrinkled with variable valleys or enamel islands. We use the term “mass” to describe this complex status, including paracone mass, metacone mass, metaconid mass, and entoconid mass. The suffixes flexus/flexid, fossette/fossettid and stria/striid are used for describing the valleys between two lophs/lophids or between two cusps. Flexus and flexid are used when the valleys are open to the tooth sides, usually in relatively unworn specimens. Stria and striid refer to the notches running down the tooth crown in buccal or lingual view. These notches are the buccal or lingual openings of the valleys. As the tooth wear deepens, the flexus or flexid will be gradually closed near the tooth sides. These closed flexus or flexids are called fossettes or fossettids. Paraflexid/fossettid/striid and metaflexid/fossettid/striid were often used for the mesial and distal valleys respectively (Stirton, 1935; Hugueney, 1975; Hugueney, 1999; Wu et al., 2004). Here we followed Lopatin (2003) by using metaflexid/fossettid/striid for the mesial valley and entoflexid/fossettid/striid for the distal flexid. We use premetafossettid instead of proparafossettid (Hugueney, 1999) or parafossettid (Lopatin, 2003) to describe the small fossa enclosed between anterolophid and metalophid I.

We developed a data matrix including 145 characters scored for 42 taxa. The 145 characters comprise 120 dental and 25 cranial characters. Marmota monax, Keramidomys fahlbuschi and Eutypomys inexpectatus were selected as outgroup taxa. Eutypomyids are widely considered as the sister group of castorids (Korth, 1994; Rybczynski, 2007; Flynn & Jacobs, 2008). Marmota and Keramidomys have the same dentition formula as that in castorids, but the phylogenetic relationship between these two taxa and castorids is probably further than that between castorids and eutypomyids. The ingroup comprises 39 taxa, of which, only Castor canadensis is an extant species. The data matrix was edited in Mesquite v3.2 software (Maddison & Maddison, 2017) and saved in the NEXUS format. The scored specimens, and the definition and arguments for the characters are listed in the NEXUS file (see Supplementary Information). Parsimony analysis was undertaken using TNT, Tree analysis using New Technology, a parsimony analysis program subsidized by the Willi Hennig Society (Goloboff, Farris & Nixon, 2008). We ran multiple replications, using sectorial searches, drifting, ratchet and fusing combined. Random sectorial search, constraint sectorial search and exclusive sectorial search were used. Ten cycles of tree drifting, 10 cycles of ratchet and 10 cycles of tree fusing were performed in the search. Default parameter settings for random sectorial search, constraint sectorial search, exclusive sectorial search, tree drifting, ratchet and fusing were used. The search level was set as 10 for 42 taxa. Optimal scores were searched with 10,000 replications. Twenty-four characters are set as “ordered” (listed in the Supplementary Information). The outgroups were not used as reference for ordering the character states. We hypothesized that the states of these characters are addable. These addable states can be observed in some chronologically succeeding castorid taxa. All characters have equal weight. We used absolute Bremer Support and relative Bremer Support (Bremer, 1994; Goloboff & Farris, 2001), calculated in TNT, to describe the stability of the phylogenetic result. TNT script for running multiple replications, using sectorial searches, drifting, ratchet and fusing combined, and script for calculating the Bremer Supports and Relative Bremer Supports were adopted from Ni et al. (2013).

Results

Systematic paleontology

Order Rodentia, Bowdich, 1821

Family Castoridae Hemprich, 1920

Genus Propalaeocastor (Borisoglebskaya, 1967)

Synonym. Asteneofiber Kretzoi, 1974: p.427; Oligotheriomys Korth, 1998: p.127

Type species. Propalaeocastor kazachstanicus (including P. habilis) (Borisoglebskaya, 1967).

Included species. P. coloradensis (Wilson, 1949b); P.  butselensis (Misonne, 1957), P. kumbulakensis (Lytschev, 1970), P. galushai (Emry, 1972), “Steneofiber aff. dehmi” (in Hugueney (1975)), P. aubekerovi (Lytschev, 1978), P. readingi (Korth, 1988), P. schokensis (Bendukidzes, 1993), P. shevyrevae (Lytschev & Shevyreva, 1994), P. sp. aff. P. shevyrevae (Lytschev & Shevyreva, 1994), P. zaissanensis (Lytschev & Shevyreva, 1994), P. primus (Korth, 1998), and P. irtyshensis Wu et al., 2004.

Distribution. Early to Late Oligocene, Eurasia; Late Eocene to Early Oligocene, North America.

Emended diagnosis. A small-sized castorid. Dental formula: 1/1, 0/0, 2/1, 3/3. Zygomatic process of maxilla forming a sloping surface. Infraorbital foramen large. Infraorbital canal short. Sciurognathous lower jaw. Digastric eminence present in some advanced species. Lower incisor enamel surface smooth, mediolaterally convex, and lacking enamel ornamentation. Lower incisor root terminating in a lateral capsule. Wide space present between lower tooth row and vertical ramus. Cheek teeth unilaterally mesodont. Upper cheek tooth crown nearly quadrate. P3 present. P4 slightly larger than M1 and M2. M3 being the smallest. Upper cheek teeth presenting complicated paracone mass and metacone mass. Premesoflexus and postmesoflexus always present. Metaflexus buccally open. p4 mesiodistally elongated. Lower molar crown rectangular. p4 larger than molars. m3 being the narrowest. Lower cheek teeth having complex metaconid mass and entoconid mass. Premesofossettid present in some species. Postmesoflexid always present. Metastylid crest present. Crown (Coronal) cementum absent.

Propalaeocastor irtyshensis Wu et al., 2004	
(Figs. 4–7; Tables 1 and 2)	

Holotype. IVPP V 13690, a right maxillary fragment preserving P4-M3. Locality XJ200203, northwest of Burqin, Xinjiang. The Irtysh Formation, Early Oligocene.

Figure 4 Maxilla and isolated upper cheek teeth referred to Propalaeocastor irtyshensis from Jeminay area, northwestern Xinjiang, China.

Yellow shadow showing the divergence of palatine nerve. (A1-3) broken maxilla with right P4-M1 (IVPP V 23138.1); (B1-3) left P4 (IVPP V 23138.2); (C1-3) left M1 (IVPP V 23138.3). (A1), (B1), (C1) lingual; (A2), (B2), (C2) buccal; (A3), (B3), (C3) occlusal. All in same scale.

Referred specimens. IVPP V 23138.1, a right maxillary fragment preserving P4-M1, IVPP V 23138.2, an isolated left P4, and IVPP V 23138.3, an isolated left M1, probably belong to the same individual; IVPP V 23139, a right dentary fragment preserving p4-m3; IVPP V 23140, a right dentary fragment preserving p4-m1; IVPP V 23141, a right dentary fragment preserving p4.

Loalities and horizon. Northeast of Jeminay County, Junggar Basin, Xinjiang (Fig. 1B). Irtysh River Formation, Early Oligocene.

Emended diagnosis. P3 present. Infraorbital foramen large, infraorbital canal short. Differing from P. kazachstanicus in having greater mandibular depth beneath p4, complete endoloph and open postmesoflexus on P4, two premesofossettids and more transverse mesoflexid on lower cheek teeth, and in lacking digastric eminence. Different from P. butselensis in having more complicated septa or spurs in buccal premesoflexus, metaflexus and premesofossettid, more distally extending mesoflexus. Different from P. kumbulakensis in having smaller size, lower tooth crown, less distally extended mesoflexus, closed postmesoflexus on P4, and two premesoflexids on p4. Differing from P. zaissanensis in having separated hypoflexus and mesoflexus on M3. Different from P. schokensis in having less massive paracone mass and metacone mass, and in lacking metalophule I on upper cheek teeth. Differing from P. aubekerovi by lacking digastric eminence and having greater mandibular depth beneath p4. Different from P. readingi in having more transversely expanded m1 and m2. Differing from P. shevyrevae in having lower tooth crown, less folded inner surface of enamel islets, and in lacking premetafossettid and having double premesofossettids on p4, and less elongated m3 lacking septum in entofossettid. Differing from P. primus in having smaller size and lower tooth crown.

Measurements. See Tables 1 and 2.

Description. The two maxillary fragments (V 13690, holotype and V 23138.1) preserve a part of the palatine process, a part of the alveolar process, and a part of the zygomatic process. The alveolar process forms the tooth sockets and holds the teeth. The dorsal side of the alveolar process is flat and smooth. It does not show any bulges for the expansion of the tooth roots. On its dorsal-medial side above the M2, it presents the opening of the caudal palatine foramen (=dorsal palatine foramen), which leads to a canal running in the maxillary-palatine suture (Figs. 4A–4B). The preserved palatine process is very small. On V 13690, only the major palatine foramen is well preserved. It is an oval and oblique opening situated between M1 and M2, and in the suture between the palatine process of the maxilla and the palatine bone (Figs. 5A–5B). On V 23138.1, the broken surface shows that the major and minor palatine foramina (=paired posterior palatine foramina) lead to short canals and meet at the caudal palatine foramen (Fig. 4A). The preserved zygomatic process of the maxilla is quite long. It extends dorsolaterally from a place at the level of the mesial root of P4. The mesial surface of the zygomatic process slopes rostrodorsally, indicating that a narrow zygomatic plate probably is present (Figs. 5A–5B). No masseteric tubercle for the superficial masseter is present on the root of the zygomatic process. Dorsal to the zygomatic process, a round and smooth surface indicates that the infraorbital foramen is probably large and round, and the infraorbital canal is very short (Figs. 5C and 5E). Dorsoventrally, the infraorbital foramen and infraorbital canal are at the level of the tooth roots, a situation as in extant protrogomorphous and sciuromorphous rodents.

Figure 5 3D virtual reconstruction of the maxillae of Propalaeocastor irtyshensis by the X-ray computed tomography.

Red shadow showing a residual P3 alveolus mesial to the mesial roots of P4; dashed cycle displaying a relative large and round infraorbital foramen dorsal to the zygomatic arch root preserved in both holotype of Burqin (A1-3: IVPP V 13690) and additional specimen of Jeminay (B1-2: IVPP V 23138.1).

On both V 13690 and V 23138.1, there is a small semi-cylindrical depression mesial to the mesial roots of P4 (Figs. 5A–5B) and (5D). This depression indicates the presence of a small single-rooted P3. Because the M3 of both specimens were already erupted and moderately worn, this small depression cannot be for the deciduous tooth. For a dP3, it should have more than one root. On the mesial surface of the P4, no obvious contacting facet is present. It is probably because the crown of P3 is very small and low, and has no tight contact with P4.

The lingual side of the upper cheek tooth crown is higher than the buccal side (Figs.  4 and 5D; Table 1). From the mesial side to the distal side of the tooth row, the tooth size decreases gradually. The lingual tooth cusps, namely protocone and hypocone, are distolingually expanded and form two fold-like structures on each tooth. The buccal cusps (paracone and metacone) and their accessory ridges form the complex paracone mass and metacone mass.

The P4 (Figs. 4A, 4B and 5A) is the largest of the upper cheek teeth. Its occlusal surface has an inverted trapezoid outline with its mesial side is wider than its distal side. The tooth can be roughly divided into four regions: the protocone region on the mesiolingual side, the paracone mass on the mesiobuccal side, the hypocone region on the distolingual side and the metacone mass on the distobuccal side. The lingual sides of the protocone region and hypocone region are separated by the deep and mesiobuccally directed hypoflexus. The buccal sides of those two regions are connected by the strong and oblique endoloph. The paracone mass is separated from the metacone mass by the deep mesoflexus. The protocone distolingual side is expanded and forms a fold-like structure. The buccal side of the protocone has two arms, the mesial protocone arm and the distal protocone arm, which merge with the anteroloph and endoloph respectively. The parastyle is a very small cusp. It is well delimited as a small node situated mesial to the paracone on a slightly worn specimens (V 23138.1-2). In the moderately worn specimen (V 13690), the parastyle is merged with the anteroloph. The paracone mass includes the paracone and two protolophs. The lingual side of the paracone smoothly extends into the protoloph I (mesial protoloph). In the less worn specimen (V 23138.1), the lingual end of the protoloph I does not join the protocone and is separated from the latter by a shallow groove. In the slightly more deeply worn specimens (V 13690, V 23138.2), the lingual end of the protoloph I connects the mesial arm of protocone through the short protolophule I. The protoloph II (distal protoloph) is a long and curved crest. Its buccal end extends to the distal side of the paracone (V 23138.1-2) or merges with paracone (V 13690). Its lingual end connects the endoloph in two speciemens (V 23138.1-2) through strong protolophule II, but is separated from the endoloph in the holotype (V 13690). On V13690, an extra fold is present at the distolingual side of protoloph II. Mesocone and mesoloph are absent. Three small fossae/flexi are present in the paracone mass: including paraflexus, lingual premesofossette and buccal premesofossette. Paraflexus is enclosed by anteroloph and protolophI. Lingual premesoflexus is enclosed by protoloph I and protoloph II. Buccal premesoflexus is enclosed by protoloph II and postparacrista. In the hypocone region, the hypocone forms a fold, which is smaller than the protocone. The mesial and distal arms of hypocone are smoothly merged with the endoloph and the posteroloph respectively. The metacone mass normally develops three ridges including double metalophs (metaloph I and metaloph II) and an extra mesial short ridge. The extra mesial ridge is the shortest, and mesiobuccally extends towards the protoloph II. The longest ridge is the metaloph I, which transversely connects the metacone and hypocone. Distal to the metaloph I, there is a long ridge referred as metaloph II here. This ridge extends distobuccally and always connects the posteroloph via a short and thin ridge (metalophule II). The postmesoflexus, which lies between the metaloph I and metaloph II, is always buccally open. The metaflexus lying in between the metaloph II and posteroloph is divided into two or three fossae by small ridges. From the buccal view, the tooth has three deep grooves, which are collectively called buccal striae. From the mesial to the distal, the three buccal striae are named as the parastria, the mesostria and the metastria. From the lingual view, the only deep groove generated by the hyoflexus is the hypostria. The hypostria is the longest. It extends nearly two-thirds of the crown height. The mesostria is the second deepest groove that reaches about a half of the crown height. The parastria and the opening of postmesoflexus are very short. P4 has a strong lingual root and two slim buccal roots.

The M1 (Figs. 4A, 4C and 5A) has a more rectangular crown than the P4, but both teeth have very similar cusp-ridge pattern. The width of M1 is larger than the length. In the paracone mass, two protolophs are present. The paraflexus and the groove between the protoloph I and protoloph II are worn into four enamel islets. The mesoflexus is a straight groove in V 23138.1 and V 23138.2. In the slightly more deeply worn specimen (V 13690) the groove is divided into two parts by a longitudinal ridge. In the metacone mass, it develops three or four ridges. The grooves between those ridges are divided into three or four fossae. As in the P4, the lingual side of M1 has one deep groove (hypostria), and the buccal side of M1 has one (mesostria) or two (mesostria and metastria) shallow grooves on moderately worn specimens (V 23138.1, .3) and lacks a groove on the deeply worn specimen (V 13690). The M1 has one strong lingual root and two slim buccal roots.

The M2 (Fig. 5A) is very similar to the M1 in both size and cusp-ridge pattern. In its paracone mass, there are two protolophs, two opened grooves and one enclosed enamel islet. The mesial groove is the long and narrow paraflexus. The distal groove is the premesoflexus. A short ridge divided the premesoflexus into an open groove on the buccal side and a small enamel islet on the lingual side. The mesoflexus is a long and curved groove separating the paracone mass and the metacone mass. In the metacone mass, there are also two metalophs. The metaloph I has an indentation in its middle part. This indentation joins the postmesofossette with the fossa between the metalophs. The metaloph II is a complete ridge that connects the metacone and hypocone. From the distolingual part of the metaloph II, it develops a short spur protruding into the metaflexus. The posteroloph is a strong ridge as a buccal extension of the distal arm of the hypocone. The development of the striae on the lingual and buccal sides of M2 is identical to those in M1 of the same dentition (V 13690).

M3 (Fig. 5A) has a narrower distal edge than in the M1-2. The hypocone of M3 is relatively small and the posteroloph is reduced. The paracone mass is almost identical to those in M1 and M2, while the metacone mass is proportionally smaller.

The newly collected specimens include three mandibular fragments (Fig. 6). Two of the three specimens preserve most of the horizontal ramus and a portion of the vertical ramus (V 23139 and V 23141). The other specimen preserves only a small part of the horizontal ramus (V 23140). The horizontal ramus of the mandible can further be divided into two parts: the part that bears the incisor and the part that bears the premolar and molars. The part that bears the incisor contains a long incisor alveolus, which runs beneath the premolar and molars and extends distally and buccally to a point lateral and above the lever of tooth crown. The tooth roots show bulges on the lingual side of the mandible, and form the alveolar juga. The buccal surface of the mandible is smooth. A large and round mental foramen is present at a place ventral to the p4 (Figs. 6C, 6F and 6I). On the mesiolingual surface of the horizontal ramus of the mandible, an oval rugose region mesioventral to the alveolus of p4 is identified as the caudoventral expansion of the mandibular symphysis (Figs. 6A and 6G). Ventral to this rugose region, no digastric eminence is present. On the ventral portion of the lingual side of the mandible, there are many nutrient foramina. A small portion of the angular process of the mandible is preserved in two specimens, and it extends caudoventrally (Fig. 6A). On the lingual side of the vertical ramus, the medial pterygoid muscle fossa is very deep. On the buccal surface of the vertical ramus, the masseteric fossa is well defined by the masseteric crest. The dorsal and ventral branches of masseteric crest are convergent nearly at a right angle, and extend to a point ventral to the m1. The coronoid process of the vertical ramus arises lateral to the m1. It includes a lateral bulge that contains the most distal extension of the incisor root. On the medial side of the coronoid process, there is a well-developed ridge (Figs. 6B and 6H). This ridge probably marks the inferior limit for the lateral pterygoid muscle. The space between the tooth row and the vertical ramus of the mandible is broad.

Figure 6 Fragmentary dentaries of Propalaeocastor irtyshensis from Jeminay, Xinjiang.

Red shadow displaying articular facet of mandibular symphysis. (A–C) fragmentary right dentary with broken p4-m3 (IVPP V 23139); (D–F) broken right dentary with p4-m1 (IVPP V 23140); (G–I) broken right dentary with p4 (IVPP V 23141). (A), (D), (G) lingual; (B), (E), (H) occlusal; (C), (F), (I) buccal. All in same scale.

Figure 7 Transverse and sagittal sections of dentaries and transverse section of lower incisor of Propalaeocastor irtyshensis of Jeminay, Xinjiang.

Showing convex enamel surface of lower incisor, permanent fourth premolar and root number of lower cheek teeth (p4:m1:m2:m3 = 2:3:3:3). (A–B) fragmentary right dentary with p4 (IVPP V 23141); (C–D) broken right mandible with p4-m1 (IVPP V 23140); (E–F) fragmentary right dentary with p4-m3 (IVPP V 23139). (G) lower incisor (IVPP V 231411). (A), (C), (E) sagittal section; (B), (D), (F), (G) transverse section.

Table 1 Measurements (in mm) of incisor and cheek teeth of Propalaeocastor irtyshensis from Jeminay, Xinjiang (L., length; W., width).

Inventory numbers	Tooth	Occlusal L. × W.	Base L. × W.	Buccal height	Lingual height	Mesostria (id) height	Hypostria (id) height	Maximum height/maximum length indices	
V 23138.1	P4	3.53 × 3.47	3.63 × 4.67	2.09	3.49	0.49	1.37	0.96	
V 23138.1	M1	3.15 × 3.79	3.15 × 4.81	1.56	2.43	–	0.84	0.77	
V 23138.2	P4	3.46 × 3.48	3.66 × 4.38	1.97	3.91	0.92	1.74	1.06	
V 23138.3	M1	3.04 × 3.16	3.27 × 3.69	1.87	3.38	–	1.64	1.03	
V 23139	p4	3.79 × 3.29	4.10 × 3.52	2.06	1.97	–	1.14	0.50	
V 23139	m1	3.12 × 3.35	3.26 × 3.62	1.39	1.35	0.2	0.29	0.43	
V 23139	m2	3.08 × 3.58	3.48 × 3.71	1.53	1.42	–	0.34	0.44	
V 23139	m3	3.05 × 3.04	3.64 × 3.22	1.44	1.45	–	0.52	0.40	
V 23140	p4	3.38 × 3.02	4.12 × 3.54	3.83	2.60	0.82	2.27	0.93	
V 23140	m1	3.05 × 3.39	3.54 × 3.98	2.49	2.07	0.49	0.9	0.70	
V 23141	i1		3.3 × 3.70					
V 23141	p4	4.31 × 4.18	4.65 × 4.41	1.83	1.54	–	0.42	0.39	

Table 2 Measurements comparison of mandibles among Propalaeocastor irtyshensis and other taxa of Propalaeocastor and Agnotocastor devius.

Asterisk numbers are re-measured from their originally illustrations i.e., Borisoglebskaya (1967), Lytschev (1970) and Lytschev & Shevyreva (1994).

Taxa	Inventory numbers	Mandibular depth beneath p4	p4-m3, mesiodistal length	p4-m2, mesiodistal length	p4-m1, mesiodistal length	p4, mesiodistal length	
P. irtyshensis	V 23139	11.8	12.4	9.2	6.2	3.79–4.10	
P. irtyshensis	V 23140	–	–	–	12.0	3.38–4.12	
P. irtyshensis	V 23141	13.2	–	–	–	4.31–4.65	
P. kazachstanicus	No. 2259-322	9.5∗	–	8.4∗	6.1∗	3.5∗	
P. aubekerovi	M − 2041∕74	9.1	11.6	9.0	6.0-7.0	3.2-3.7	
P. schokensis	No. 15/48	–	15.7	–	–	–	
P. schokensis	No. 15/48	–	16.4	–	–	–	
A. devius	No. 3463-4	7.0∗	10.0∗	7.7∗	5.1∗	3.2∗	
P. kumbulakensis	M − 2020∕66	11.4∗	19.8∗	16.1∗	10.9∗	6.3∗	
P. readingi	CSC 80-1	11.0	–	10.6		3.35	
P. coloradensis	UCM 19809	14.6	–	11.9		4.1	
P. galushai	FAM 79310	10.1	11.7	9.1	–	3.4	

The lower incisor is only preserved in one specimen (V 23141). The cross-section of this lower incisor is in a rounded triangular shape. The pulp cavity is large and round. The enamel band of the incisor is smooth and buccoventrally convex (Fig. 7G).

The buccal sides of the lower cheek tooth crowns are slightly higher than the lingual crown side (Table 1). From p4 to m3, the sizes are gradually reduced. On all the cheek teeth, the protoconid and hypoconid are large and mesiobuccally protruding. The metaconid and entoconid and the ridges associated with them from the complicated metaconid mass and entoconid mass.

All the three mandibles preserve the fourth premolar (Figs. 6 and 7). The crown of the p4 (Figs. 6B, 6E and 6H) has a trapezoid outline with its mesial side narrower than its distal side. The hypoflexid and mesoflexid (=mesofossettid when its lingual side is closed) form a waist that divides the tooth crown into mesial and distal lobes. The protoconid, the anterolophid and the mesial part of the ectolophid are merged into a strong curved ridge that defines the buccal margin of mesial lobe. The metaconid, the lingual part of the metalophid II, the metastylid and the metastylid crest are fused into another curved ridge that forms the lingual margin of the mesial lobe. In less worn individuals (V 23139, 23140), the cusps and ridges in the mesial lobe enclose three fossae (Figs. 6B and 6E). In a deeply worn individual (V 23141), only one fossa is left (Fig. 6H). The metaflexid (=metafossettid when its lingual side is closed), which is enclosed between the anterolophid and metalophid II, is a long and curved groove (or fossa). The metalophid I is present as a spur derived from the anterolophid and extends into the metafossettid. Between the metalophid II and the metastylid crest, two fossae are present, namely the buccal premesofossettid and the lingual premesofossettid (Fig. 6B and 6E). The distal side of the lingual premesofossettid is open in one specimen (Fig. 6E). The two fossae disappear in the heavily worn specimen (V 23141, Fig. 6H). The mesoflexid is a long and deep groove that extends transversely across more than half of the crown width. The hypoflexid on the buccal tooth side has a broad opening. It extends distolingually to the mesiolingual side of the hypoconid. The hypoflexid and mesoflexid are separated by the ectolophid. The ectolophid also connects the mesial and distal lobes. The distal lobe is formed by the hypoconid, entoconid and the ridges and arms associated with those two cusps. The hypoconid is very large and forms the buccal half of the distal lobe. The posterolophid, the entoconid, the hypolophids and the distal part of ectolophid form the lingual half of the distal lobe. The mesial hypolophid (hypolophid I) and the small postmesofossettid are present in the less worn specimen (Fig. 6E). The entoflexid is present as long groove between the distal hypolophid (hypolophid II) and posterolophid. Complicated enamel folds developed from the hypolophid II and posterolophid protrude into the entoflexid. In the deeply worn specimen (V 23141), these folds connect to each other and divide the entoflexid into 3 enamel islets. Two broad roots are present on p4 (Fig. 7B, 7D and 7F).

The m1 (Figs. 6B and 6E) is preserved on two specimens (V 23139, V 23140). Both of them are heavily worn. The m1 has a rectangular crown, with its width larger than its length. As in the p4, the conspicuously deep mesoflexid and hypoflexid form a waist and divide the tooth into mesial and distal lobes. The protoconid, metaconid and the ridges associated with the mesial lobe tend to merge together. One or two enamel islets are enclosed in the mesial lobe. It is hard to deduce whether they homologize with the metafossettid or with the premesofossettid. The mesoflexid is lingually open on V 23140 but closed on V 23139. The hypoflexid of m1 is narrower than that of the p4. The distal lobe of m1 is slightly broader than the mesial lobe. In the slightly worn specimen (V 23140), a small enamel islet is identified as the postmesofossettid. A transverse curved groove is the entoflexid. In the heavily worn specimen (V 23139), the hypoconid, the entoconid, the hypolophids, and the posterolophid completely merge. The tooth has three roots, including two slim mesial roots and one broad distal root (Figs. 7B, 7D and 7F).

Only one specimen (V 23139) preserves m2 and m3. The m2 (Fig. 6B) is very similar to the m1. A shallow oval fossa in the middle of mesial lobe can be identified as the metafossettid. The mesoflexid between the mesial lobe and distal lobe is lingually closed. In the distal lobe, the transverse fossa is identified as the entofossettid. As in m1, m2 also has two slim mesial roots and one broad distal (Figs. 7B and 7F).

The m3 (Fig. 6B) is very similar to m1 and m2, but is slightly longer and narrower. Its mesial lobe has two fossae. The large buccal one is identified as the metafossettid. The tiny lingual fossa is identified as the premesofossettid. As in m2, the mesoflexid is lingually closed. In the distal lobe, the large and oblique entoflexid is preserved. The tooth has three roots as those of m1 and m2 (Fig. 7F).

Phylogenetic analysis

The parsimony search of our phylogenetic analysis provided 6 most parsimonious trees. Each has a best score of 543 steps (CI = 0.3554 and RI = 0.6625). The majority-rule consensus shows that most clades have 100% consensus (Fig. 8). The inner group (castorids) is a monophyletic group with robust absolute and relative Bremer Supports. Character-state optimization using the Accelerated transformation (ACCTRAN) criterion shows that the inner group is supported by 21 dental and two cranial synapomorphies (Table 3). Six species of Propalaeocastor (P. schokensis, P. butselensis, P. kazachstanicus, P. kumbulakensis, P. irtyshensis, P.  shevyrevae), four species previously referred to Agnotocastor (P.  galushai, P. readingi, P. coloradensis, P.  aubekerovi), P. primus and “Steneofiber aff. S. dehmi” form a monophyletic group. The absolute and relative Bremer Supports show that the monophyly of this group is quite robust. This result supports our systematic revision of Propalaeocastor. Character-state optimization shows that Propalaeocastor clade is supported by seven dental and two cranial synapomorphies (Table 4).

Figure 8 Majority-rule consensus of six most parsimonious trees.

Parsimony analysis is based on a data matrix including 145 characters scored for 42 taxa. Marmota monax, Keramidomys fahlbuschi and Eutypomys inexpectatus were selected as outgroup taxa. Numbers before the slashes at the internodes are the absolute Bremer Support values; numbers after the slashes are Relative Bremer Support values; numbers after the comma are percentage of consensus. Internodes without the percentage of consensus show the topologies that are present in all the six most parsimonious trees. The geographic distribution of all the taxa was mapped on the majority-rule consensus tree and the ancestral states were reconstructed using the parsimony criterion in Mesquite 3.2 (Maddison & Maddison, 2017). Red clades represent Asian origin, blue clades represent European origin, and black clades represent North American origin. Clades in dashed line indicate equally-parsimonious or ambiguous Asian, European or North American origins. Scale bar equals 20 character changes.

Table 3 Synapomorphy list for Propalaeocastor.

Tree description were undertaken in PAUP* Version 4.0b10. The majority-rule consensus tree was rooted using outgroup method. Character-state optimization is done under Accelerated Transformation (ACCTRAN) model. The double arrow “==>” represents unambiguous changes, and the single arrow “−−>” represents ambiguous changes.

Character No.	Characters	CI	State changes	
2	Upper P4 size relative to upper M1	0.200	Subequal	==>	Large	
33	Upper P4 postmesofossette shape in metacone mass in medium wear stage	0.250	Transversely long valley	==>	Small enamel island-like fossa(e)	
54	Upper P4 postcingulum buccal end fused with the metacone in moderate wear	0.333	Present, the metaflexus is closed	−−>	Absent, the metaflexus is open	
55	Upper M1-2 postcingulum buccal end fused with the metacone in moderate wear	0.333	Present, the metaflexus is closed	−−>	Absent, the metaflexus is open	
79	Lower p4 metastylid crest	0.250	Absent	==>	Present	
106	Lower m1-2 metastylid crest	0.200	Absent	==>	Present	
108	Lower m1-2 premesofossettid	0.250	Absent	==>	Present	
122	Korth (2001) posterior palatine foramina in palatine-maxillary suture	0.250	No	−−>	Yes	
145	Mandible, space between the lower tooth row and vertical ramus	0.143	Narrow	−−>	Broad	

Table 4 Synapomorphy list for castorids.

Tree description were undertaken in PAUP* Version 4.0b10. The majority-rule consensus tree was rooted using outgroup method. Character-state optimization is done under Accelerated Transformation (ACCTRAN) model. The double arrow “==>” represents unambiguous changes, and the single arrow “––>” represents ambiguous changes.

Character No.	Characters	CI	State changes	
11	Cheek teeth crown height	0.333	Brachydont	==>	Mesodont	
12	Upper teeth lingual higher than buccal or lower teeth buccal higher than lingual	0.500	Absent	−−>	Present	
13	Cheek teeth crown structure	0.500	Bunodont-lophodont	−−>	Lophodont	
16	Upper incisor buccal surface flatness	0.250	Very convex	−−>	Slightly convex	
17	Lower incisor buccal surface flatness	0.250	Very convex	==>	Slightly convex	
44	Upper M1-2 preprotocrista buccal end height relative to the paracone mass or paracone	0.200	Lower	−−>	Subequal	
46	Upper M1-2 preprotocrista buccal end (parastyle) fused with the paracone mass or paracone	0.250	Absent	==>	Present	
65	Upper cheek teeth mesocone	0.500	Present	−−>	Absent	
66	Upper cheek teeth mesoloph	0.500	Present	−−>	Absent	
67	Upper cheek teeth mesostyle	1.000	Present	==>	Absent	
68	Lower p4 size relatvie to m1	0.250	Smaller	−−>	Larger	
69	Lower p4 anteroconid	1.000	Present	==>	Absent	
70	Lower p4 anterolophid (=paracristid)	0.333	Absent	−−>	Present	
71	Lower p4 postprotocristid	0.500	Absent	−−>	Present	
82	Lower p4 mesoconid	0.500	Present	−−>	Absent	
83	Lower p4 mesolophid	0.500	Present	−−>	Absent	
93	Lower cheek teeth hypoflexid extension	0.667	Shallow	−−>	Medium	
95	Lower m1-2 anterior cingulid	0.500	Present	−−>	Absent	
103	Lower m1-2 premetafossettid enclosing by the metalophid I and the paracristid (anterolophid)	0.167	absent	==>	Present	
109	Lower m1-2 mesoconid	0.500	Present	−−>	Absent	
110	Lower m1-2 mesolophid	0.500	Present	−−>	Absent	
130	Xu (1995) C2. digastric eminence on mandible	0.200	Absent	−−>	Present	
142	Mandible capsular process	0.333	Weak	==>	Large	

Discussion

Comparisons. Many researchers suggested that Propalaeocastor is similar to Steneofiber (Lytschev, 1970; Lytschev & Shevyreva, 1994; Lopatin, 2003; Lopatin, 2004; Bendukidze, De Bruijn & Van Den Hoek Ostende, 2009). Wu et al. (2004) also listed seven characters shared by the two genera. Steneofiber was established by Geoffroy Saint-Hilaire (1833) for the beavers fossils discovered at Langy (Allier) in the basin of Saint-Gérand-le-Puy, France. Its type species is S. castorinus identified by Pomel (1846) (see Stirton, 1935). The new Jeminay specimens reported here show that Propalaeocastor differs from Steneofiber by presenting a P3, and in having a larger P4 and p4 relative to the molars, a mesiodistally more elongated P4 and p4, relatively wider molars, and more complicated ridge-fossa pattern. In Propalaeocastor, the metalophs on the upper teeth and the hypolophids on the lower teeth are divided to two or three branches. The upper teeth and the lower teeth usually have a premesofossette and postmesofossette, and a premesofossettid and postmesofossettid respectively. The mesoflexus and mesoflexid are more transversely orientated. In the narrower flexures and fossae of Propalaeocastor, many crenulated enamel folds usually develop from the adjacent lophs or ridges. In a sharp contrast, Steneofiber has a relatively much simpler and less crenulated ridge-fossa pattern.

“Steneofiber aff. dehmi” from the early Oligocene Saint-Martin-de-Castillon of France (Hugueney, 1975) was treated as a member of Propalaeocastor by Wu et al. (2004). Here we follow their assignment. As in other Propalaeocastor specimens, “Steneofiber aff. dehmi” has premesofossettes and postmesofossettes on the upper cheek teeth, and has premesofossettids and postmesofossettids on the lower cheek teeth. Compared to P. irtyshensis, “Steneofiber aff. dehmi” is larger. The mesoflexus on the upper cheek teeth are more distally extended due to lacking a metalophule I. The lower cheek teeth are more slender and have the metastylid crests.

Propalaeocastor shares many similarities with the North American late Eocene to early Oligocene Agnotocastor, which is widely regarded as the oldest castorid genus (Korth, 1994; Xu, 1995; Xu, 1996; Flynn & Jacobs, 2008). As in Propalaeocastor, a single-rooted P3 is also present in Agnotocastor. Previously six species were included in this genus. Four of them, namely the type species A. praetereadens, “A.” coloradensis, “A.” galushai and “A.” readingi, are from North America. Two species, “A.” aubekerovi and A. devius, are from Kazakhstan of Asia (Stirton, 1935; Wilson, 1949b; Emry, 1972; Lytschev, 1978; Korth, 1988; Lytschev & Shevyreva, 1994). Based on the dental morphology and our phylogenetic analysis, we transfer four species (coloradensis, galushai, readingi and aubekerovi) to Propalaeocastor, and reserve only A. praetereadens and A. devius, in Agnotocastor. A. praetereadens is from the White River Formation of South Dakota, USA, and is represented by a skull (AMNH 1428). As in P. irtyshensis, P3 is also present in A. praetereadens and A. devius. A. praetereadens differs from P. irtyshensis in having simpler dental morphology that lacks premesofossettes and postmesofossettes on upper cheek teeth. A. devius from Mayliaby of Zaissan Basin (Lytschev & Shevyreva, 1994) also has a distinctly simpler dental morphology. It differs from P. irtyshensis in having smaller tooth size, shallower mandibular depth beneath the p4, and more caudodorsally extending angular process of the mandible.

P. coloradensis, P. galushai, and P. aubekerovi include only lower jaw fragments and lower teeth. They all have distinct postmesofossettids on their lower cheek teeth. This is the diagnostic feature of Propalaeocastor. Furthermore, the position of the mental foramen of these three species is also beneath the anterior root of p4. P. readingi from the Orella Memmber of Brule Formation of Dawes County in Nebraska was named based on a mandibular fragment preserving p4-m2 (CSC 80-1; Korth, 1988). Later, Korth (1996a) described additional specimens of this species and emended its diagnostic features. Its dental morphology displays a complicated pattern, such as presenting the premesofossette and postmesofossette on the upper cheek teeth, and the postmesofossettid on the lower cheek teeth. These features are typically seen in Propalaeocastor.

P. coloradensis from the Brule Formation of Loagan County in Colorado (Wilson, 1949b) differs from P. irtyshensis in having greater tooth size, lower tooth crown, deeper mandibular depth beneath p4 (Table 2), and in presenting a digastric eminence and distinct metastylid crests on the lower cheek teeth. P. galushai from the South Fork of Lone Tree Gulch in Wyoming (Emry, 1972) is similar to P. irtyshensis in size (Table 2). P. galushai has a stronger digastric eminence and lower tooth crowns. Its p4 metaconid mass and entoconid mass show weaker connections to the protoconid and the hypoconid respectively than in P. irtyshensis. P. readingi is slightly larger than P. irtyshensis (Table 2). P. irtyshensis differs from P. readingi in having more transversely expanded m1 and m2. Given the very wide geographic separation, the minor difference between P. readingi and P. irtyshensis is remarkable. Compared to P. aubekerovi from Tort-Molla, Ulutau, Dzhezkazgan Province in Kazakhstan (Lytschev, 1978), P. irtyshensis is different by lacking the digastric eminence and presenting much thicker mandibular depth beneath p4 (Table 2).

Propalaeocastor primus from the Brule Formation of Fitterer Ranch in North Dakota, USA was raised as the type species of Oligotheriomys (Korth, 1998). Here we take Oligotheriomys as the junior synonym of Propalaeocastor. P. primus has only one right maxilla preserving M1-2 (FAM 64016). The preserved alveolus indicates that the P3 is present. The molar morphology of this species is complicated. As in other species of Propalaeocastor but different from other basal castorids, the paracone and metacone and the ridges associated with these two cusps form complex paracone mass and metacone mass. The premesofossette and postmesofossette are clearly present. P. primus differs from P. irtyshensis by its distinctly larger size, higher crown and much shallower hypoflexus and mesoflexus.

The type species Propaleocastor kazachstanicus was discovered from Kyzylkak, Dzhezkazgan and Kazakhstan (Borisoglebskaya, 1967). Compared to P. kazachstanicus, P. irtyshensis has a relatively deeper mandibular depth beneath the p4 (Table 2). Caudoventral to the mandibular symphysis, a small digastric eminence is present in P. kazachstanicus, but not in P. irtyshensis. The preserved part of the angular process in P. irtyshensis shows that the angular process probably is more caudoventrally directing than that in P. kazachstanicus. P. irtyshensis has more transverse mesoflexids on the lower cheek teeth than those in P. kazachstanicus. (Lytschev & Shevyreva, 1994) referred nine isolated cheek teeth discovered from Maylibay of Zaissan Basin to P. kazachstanicus (Fig. 2 in Lytschev & Shevyreva, 1994). These teeth differ from P. irtyshensis by having narrower crowns, and by having more distally extended mesoflexus on M1-2 and only one premesofossettid on p4.

Compared to P. butselensis from the Hoogbustsel-Hoeleden in Belgium (Misonne, 1957), P. irtyshensis has a more complicated dental structure. The premesofossette, metaflexus and premesofossettid in P. irtyshensis are usually divided by extra septa or spurs. The mesoflexus in P. irtyshensis is more distally extending, while in P. butselensis it is nearly transverse. “Steneofiber cf. S. butselensis” from the Buran Svita of Podorozhnik, locality K15, south of Lake Zaissan (Emry et al., 1998) was also regarded as a member of Propalaeocastor by Wu et al. (2004). These specimens are very similar to P. irtyshensis. They have a slightly smaller tooth size and relatively narrower m1-2 than P. irtyshensis.

P. kumbulakensis was discovered from the Kumbulak cliffs, the loc. Altyn Schokysu, the loc. Akotau, the loc. Akespe, and the loc. Sayaken near the Aral Sea (Lytschev, 1970; Lopatin, 2003; Lopatin, 2004; Bendukidze, De Bruijn & Van Den Hoek Ostende, 2009). It is much larger and more robust than P. irtyshensis. The upper teeth of P. kumbulakensis have premesofossettes, postmesofossettes and double metalophs. The lower teeth have the postmesofossettids and double hypolophids. These features are similar to those in P.  irtyshensis. Similar to P. irtyshensis, P. kumbulakensis does not have a digastric eminence, and its angular process extends caudoventrally. The p4 of P. kumbulakensis has a single premesofossettid, and a large groove merged by mesoflexid and metaflexid. The hypoflexid in P. kumbulakensis is very deep and extends lingually on the p4-m1. The postmesofossettid is absent on the p4, but is present on the m1. The tooth crown of the m1 in P. irtyshensis is mesial-distally more compressed and buccal-lingually wider than in P. kumbulakensis.

P. schokensis from the Altyn Schokysu of Kazakhstan (Bendukidze, 1993) is larger than P. irtyshensis (Table 2). It differs from P. irtyshensis in having much more massive paracone and metacone masses on upper cheek teeth but with simpler metaconid mass on the p4 (see Bendukidze, De Bruijn & Van Den Hoek Ostende, 2009).

Compared to P. irtyshensis, P. shevyrevae from Talagay in the Zaissan Basin (Lytschev & Shevyreva, 1994) has relatively lower tooth crowns, less folded inner surfaces of enamel islets, smaller p4 with a more rounded protoconid and a less projected hypoconid. The lower cheek teeth of P. shevyrevae have premetafossettids and single premesofossettids. The m3 is more elongated and has two metafossettids. Propalaeocastor aff. P. shevyrevae from the Podorozhnik and the Novei Podorozhnik in the Zaissan Basin (Lytschev & Shevyreva, 1994) is similar to P. irtyshensis in overall morphology. The P4 of Propalaeocastor aff. P. shevyrevae is slightly larger and more slender than that of P. irtyshensis. It differs from P. irtyshensis in having more tortuous enamel folds that protrude into the fossae on upper teeth, and in having one premesofossettid on p4.

P. zaissanensis from the Talagay in the Zaissan Basin (Lytschev & Shevyreva, 1994) is very close to P. irtyshensis in both tooth size and morphology. P. zaissanensis differs from P. irtyshensis in having a relatively narrow p4, and a hypoflexus transversely confluent with the mesoflexus on M3.

Some other basal castorid genera including Miotheriomys, Microtheriomys, Minocastor and Neatocastor were regarded as close relatives of Propalaeocastor (Korth, 1996b; Korth, 2004; Korth & Samuels, 2015; Mörs, Tomida & Kalthoff, 2016). All these genera include their type species only. Korth (1996b) dumped “Steneofiber” hesperus (Douglass, 1901), “S.” complexus (Douglass, 1901) and “S.” montanus Scott, 1893 into one species (“S.” hesperus) and established a new genus (Neatocastor) for it. The type specimens of Neatocastor hesperus was from the Arikareean (late Oligocene) of the Blacktail Deer Creek of Beaverhead County in Montana. It has a dP3 and relatively complicate upper dental morphology, but with relatively simple lower teeth similar to that of Steneofiber. N. hesperus differs from Propalaeocastor in having more convex lower incisor enamel surface and weakly developed endolophs on the upper cheek teeth, and in lacking the postmesofossettes on the upper cheek teeth and the premesofossettids and the postmesofossettids on the lower cheek teeth. Miotheriomys stenodon is from the Runningwater Formation (Early Hemingfordian, Early Miocene) of western Nebraska (Korth, 2004). It differs from Propalaeocastor in lacking the premesofossettids and the postmesofossettids on the lower cheek teeth. Microtheriomys brevirhinus is from the John Day Formation (early Early Arikareean, late Early Oligocene) in Oregon (Korth & Samuels, 2015). It is different from Propalaeocastor by lacking the P3, lacking the premesofossettids and the postmesofossettids on the lower cheek teeth, and presenting the dorsal palatine foramen entirely within the palatine bone. Minocastor godai is from the lower Miocene of the Kani Basin in central Japan (Mörs, Tomida & Kalthoff, 2016). It is distinctly larger than the all the species of Propalaeocastor. The enamel surface of its lower incisor is more convex than that of Propalaeocastor. Its lower cheek teeth are more Steneofiber-like by presenting very reduced presmesofossettids and postmesofossettids. Its upper cheek teeth display a relatively complicated dental pattern as in Propalaeocastor, but without the postmesofossette.

The new Propalaeocastor irtyshensis specimens reported here show that the dental morphology of this species is similar to other early castorids, such as Agnotocastor, and Neatocastor and Microtheriomys. On the other hand, P. irtyshensis also possesses some features that superficially similar to the eutypomyids, such as two upper premolars and complicate cusp and ridge patterns. Among castorids, it is known that two upper premolars are present in Agnotocastor devius (Lytschev & Shevyreva, 1994), and some North American early castorids, such as Agnotocastor, Neatocastor and “Oligotheriomys” (which is sunk into Propalaeocastor here) of North America (Stirton, 1935; Korth, 1996b; Korth, 1998).

Extant and fossil castorid skulls clearly exhibit the sciuromorphous skull pattern, while the sister-group of castorids, the eutypomyids, show the protrogomorphous morphology (Wood, 1965). In basal castorids, it was not clear whether they have the protrogomorphous pattern or the sciuromorphous pattern. The zygomatic process of maxilla of P. irtyshensis displays a conspicuous mesiodorsally-distoventrally oblique surface. In protrogomorph skulls the zygomatic root ventral to the infraorbital foramen has an oval roughened scar for the attachment of the anterior part of the deep masseter and the superficial masseter. No such a scar is present in P. irtyshensis. A sloping zygomatic process of maxilla without the oval scar indicates that a rudimentary sciuromorph-like zygomatic plate probably is present (Figs. 5A–5B). Medial to this rudimentary zygomatic plate and dorsal to the zygomatic root of the maxilla, it presents a smooth and round surface. This surface indicates that the infraorbital foramen is large and round, and the infraorbital canal is short. The rudimentarily developed zygomatic plate coupled with a large infraorbital foramen and canal probably is the plesiomorphic feature for all castorids. In extant beavers, the infraorbital foramen is small, the infraorbital canal is long, and the sciuromorph zygomatic plate forms a deep fossa locating lateral to the infraorbital canal (Cox & Baverstock, 2016). More derived fossil beavers, such as Monosaulax, Eucastor, Procastoroides etc., all have the sciuromorph-like zygomatic plate with a deep fossa. In myomorphous rodents, the zygomatic plate is present and the infraorbital foramen is large. Different from the protrogomorphous rodents, the large infraorbital foramen in the myomorphous rodents is mediolaterally compressed. The large infraorbital foramen in P. irtyshensis does not show any sign of compression.

Xu (1996) argued that Propalaeocastor kumbulakensis should be assigned to the eutypomyid genus Eutypomys because the lower jaw of P. kumbulakensis does not have a digastric eminence, and its angular process extends caudoventrally. We re-examined the mandibular specimens of Propalaeocastor and found that the digastric eminence is variably present in different species. In P. kumbulakensis, P. irtyshensis, P. readingi and P. devius, the digastric eminence is absent, while in some other species, such as P. coloradensis, P. galushai, P. aubekerovi and P. kazachstanicus it is well-developed. In P. irtyshensis, the articular facet of the mandibular symphysis has a large expansion beneath the genial fossa. The presence of this enlargement strengthens the mandibular symphysis. In all the castorids with genial region preserved, the articular facet of the mandibular symphysis all has this ventral expansion. When the digastric eminence is present, the articular facet always extends onto it. The so-called digastric eminence probably is a part of articular expansion related to the strengthening of the mandibular symphysis, not just for providing the arising places for the digastric muscles. In that sense, the expansion of the articular facet of the mandibular symphysis is associated with the appearance of digastric eminence and therefore should be regarded as a feature shared by all castorids.

The angular process of mandible is also variably present in different species of Propalaeocastor and other basal castorids. In some species preserving that part, such as P. kumbulakensis, P. irtyshensis, P. aubekerovi, and P. galushai, the angular process of the mandible extends caudoventrally, while in P. kazachstanicus, the angular process shows a tendency of caudodorsal extension (Fig. 9). It is likely that the direction of the angular process is related to the development of the medial pterygoid muscle, and probably also superficial masseter.

Figure 9 Chronologic and geographic distribution of Propalaeocastor and Agnotocastor, and comparisons of dentary and dental patterns.

Displaying the developments of digastric eminence and angular process of the mandible extending orientations of their mandibles and similarities of dental constructions. Asterisk showing the type species of Agnotocastor and Propalaeocastor. Except for the figures of Propalaeocastor irtyshensis (dentary, IVPP V 23141; lower dentition, IVPP V 23139, upper dentition, IVPP V 13690), the illustrations in the Dentaries, Lower Dentitions and Upper Dentitions columns are facsimiles of their original figures (Stirton, 1935; Wilson, 1949b; Borisoglebskaya, 1967; Lytschev, 1970; Emry, 1972; Hugueney, 1975; Lytschev, 1978; Korth, 1998; Korth, 1996a; Korth, 1998; Lytschev & Shevyreva, 1994). Abbreviations used in left column are biochrons of North American Land-Mammal Ages (NALMA): Ch-1 = Early Chadronian; Ch2-3 = Middle Chadronian; Ch4 = Late Chadronian; Or1–Or4 = Orellan; Wh1–Wh2 = Whitneyan (see Flynn & Jacobs, 2008). Dentaries and dentitions are in same scales, respectively.

Korth (1994), Rybczynski (2007) and Flynn & Jacobs (2008) enumerated many features of Castoridae that are different from Eutypomidae, such as the relatively high rostrum cross-sectional shape, wider nasals, the small and mediolaterally compressed infraorbital foramen, the long infraorbital canals, the distinctive chin process (symphyseal flange, or mandibular eminence), and the base of lower incisor terminating in a lateral bulbous expansion etc. Xu (1996) once defined the castorids as “the rodents that have sciuromorphous masseter arrangement on the skull and a derived mandible here termed the beaver-pattern mandible”. His “beaver-pattern mandible” is referred to a mandible presenting “digastric eminence” and “the angle extending up posteriorly”. Eutypomyidae is characterized by presenting a narrow zygomatic plate, a large and round infraorbital foramen, a short infraorbital canal, two upper premolars, and a lower jaw lacking the digastric eminence and having a caudoventrally extending angular process (e.g., Wahlert, 1977; Korth, 1994). Obviously all these features are cranial features. Our phylogenetic analysis is based on a data matrix that includes mainly dental features. It is not possible to evaluate all the differences between castorids and eutypomyids mentioned above, but our analysis does discover that presence of digastric eminence and large capsular process are synapomorphies of all castorids. The caudal palatine foramen situated in maxillary-palatine suture is a feature generally accepted as a diagnostic character for castorids (Korth, 2001). In our analysis, this feature is a synapomorphy of Propalaeocastor, but not the synapomorphy of castorids (Tables 3 and 4). On the other hand, our phylogenetic analysis suggests that dental features are also important for distinguishing the castorids and eutypomyids. Twenty-one dental features are synapomorphies of the castorid clade (Table 4).

Phylogeny and applications. It is generally believed that Agnotocastor and Propalaeocastor are close to each other (Korth, 2001; Korth, 2004; Korth & Samuels, 2015; Mörs, Tomida & Kalthoff, 2016). Our phylogenetic analysis suggests that some species of “Agnotocastor”, namely of P. galushai, P. readingi, P. coloradensis and P. aubekerovi, should be reassigned to Propalaeocastor. “Steneofiber aff. S. dehmi” from the Saint-Martin-de-Castillon in France (Hugueney, 1975) is morphologically more similar to Propalaeocastor than to Steneofiber. Wu et al. (2004) assigned this species to Propalaeocastor but did not give a new name to it. The result of our analysis indicates that “Steneofiber aff. S. dehmi” and three North American species (P. galushai, P. readingi and P. primus) form a monophyletic group. This result is consistent with our comparisons and that of Wu et al. (2004). P. primus was described as a new species based on the comparison with Anchitheriomys (Korth, 1998). Our result suggests that P. primus is the sister group of P. readingi, deeply nesting in the monophyletic clade of Propalaeocastor. To keep the monophyly of Propalaeocastor, we should sink Oligotheriomys to Propalaeocastor. The type species of Agnotocastor (A. praetereadens) and A. devius (Stirton, 1935; Lytschev & Shevyreva, 1994) form a monophyletic group with high Bremer Support. They are not the sister group of Propalaeocastor, but stem taxa that eventually lead to the crown castoroid group.

Steneofiber was suggested to be very close to Propalaeocastor. Lytschev & Shevyreva (1994), Lopatin (2003) and Lopatin (2004) even suggest that Propalaeocastor is a junior synonym of Steneofiber. Some species, such as P. butselensis, P. kumbulakensis and P. schokensis, were referred to Steneofiber (Hugueney, 1975; Lytschev & Shevyreva, 1994; Lopatin, 2003; Lopatin, 2004), while Wu et al. (2004) and Bendukidze, De Bruijn & Van Den Hoek Ostende (2009) referred them to Propalaeocastor. Our phylogenetic analysis indicates that Steneofiber is a polyphyletic group. The type species, Steneofiber castorinus, is the sister group of Chalicomys + Castor, suggesting that Steneofiber is far more derived than the basal castorid Propalaeocator.

Korth (2001) believed that Propalaeocastor is close to Oligotheriomys and Anchitheriomys, and assigned these genera to the Tribe Anchitheriomyini of the Subfamily Agnotocastorinae. Later, Korth (2004) named Miotheriomys and elevated the Tribe Anchitheriomyini into the Subfamily Anchitheriomyinae to include Propalaeocastor, Oligotheriomy, Anchitheriomys and Miotheriomys. Korth & Samuels (2015) named Microtheriomys and also include it into the Subfamily Anchitheriomyinae. Mörs, Tomida & Kalthoff (2016) named Minocastor and raised a tribe (Tribe Minocastorini) of the Subfamily Anchitheriomyinae to include Minocastor, Microtheriomys, Miotheriomys, Oligotheriomys and Propalaeocastor. Our phylogenetic analysis indicates that Oligotheriomys is nested in the species of Propalaeocastor, and we synonymize Oligotheriomy to Propalaeocastor to reflect this result. In our phylogenetic analysis, we discovered that Anchitheriomys, Minocastor and Miotheriomys are close to each other, but form a paraphyletic group. Microtheriomys takes a more basal position than those three genera.

The late Eocene Propalaeocastor galushai is the oldest-known castorid. It possesses many plesiomorphic features, such as the persistence of P3, the angular process of the mandible extending caudoventrally, and the complicate dental pattern. These features are present in most of the species of Propalaeocastor, and they are also present in the eutypomyids, which are widely considered as the sister group of castorids. Therefore, these features are likely plesiomorphic for all castorids. However, our phylogenetic analysis shows that P. galushai is not the most basal castorid, not even the most basal Propalaeocastor (Fig. 8). This result would suggest that the diversification of Propalaeocastor is before the late Eocene.

It was suggested that castorids originated in North America, and probably dispersed into Asia during the Early Oligocene (Lytschev, 1978; Lytschev & Shevyreva, 1994; Xu, 1995; Korth, 2001; Rybczynski, 2007). This hypothesis is supported by our phylogenetic analysis. The place of origin of Propalaeocastor is uncertain. Based on the result of our phylogenetic analysis, it is equally parsimonious to predict an Asian origin, a European origin or a North American origin of castorids (Fig. 8). A castorid earlier than P. galushai and more primitive than P. irtyshensis and P. butselensis is yet to be discovered. The rapid radiation of castorids in the early Oligocene probably is propelled by the global climate changes during the Eocene-Oligocene transition (EOT). Dramatic sea level drop during the EOT probably produced multiple passages enabling the dispersal of Propalaeocastor-like basal castorids across the northern continents. These basal castorids then quickly became diversified and adaptive to new ecological niches.

Supplemental Information

Supplemental Information 1 Data matrix used for phylogenetic analysis

The data matrix was edited in Mesquite v3.2 software (Maddison & Maddison, 2017) and saved in the NEXUS format.

Click here for additional data file.

Supplemental Information 2 Relatived references to the taxa in the data matrix for phylogenetic analysis

Click here for additional data file.

We thank the field participants in the Jeminay and Burqin regions of Xinjiang, especially to Mr. Shaoguang Zhang and the local drivers Messrs. Ming Hao, Hongwei Li and Jianxun Yan. We would like to express our gratitude to Mr. Yongchun Yang for his enthusiastic favors in our fieldworks. Many thanks are also due to Mr. Yemao Hou for the computedtomography. Our deepest gratitude goes to the Dr. Joshua X. Samuels and the anonymous reviewer for their careful work and thoughtful suggestions that have helped improve this paper substantially.

Abbreviations

AMNH American Museum of Natural History

CSC Chadron State College

FAM Frick American Mammals, Department of Vertebrate Paleontology, the American Museum of Natural History

IVPP Institute of Vertebrate Paleontology and Paleoanthropology, Chinese Academy of Sciences

UCM University of Colorado Museum

XJ prefix to Xijiang, field localities of the IVPP

Additional Information and Declarations

Competing Interests

Author Contributions

Data Availability

The authors declare there are no competing interests.

Lüzhou Li performed the experiments, analyzed the data, contributed reagents/materials/analysis tools, wrote the paper, prepared figures and/or tables, reviewed drafts of the paper.

Qiang Li and Xijun Ni conceived and designed the experiments, performed the experiments, analyzed the data, contributed reagents/materials/analysis tools, wrote the paper, prepared figures and/or tables, reviewed drafts of the paper.

Xiaoyu Lu contributed reagents/materials/analysis tools, reviewed drafts of the paper.

The following information was supplied regarding data availability:

The raw data has been supplied as a Supplementary File.

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
