# Peer review of "Morphology of an Early Oligocene beaver Propalaeocastor irtyshensis and the status of the genus Propalaeocastor"

_PeerJ, doi:10.7717/peerj.3311_

## Round 0.1 · original submission · Major Revisions

· Academic Editor

Major Revisions

I am in agreement with the reviewers that this is an article worthy of inclusion in PeerJ. However, it needs some revision before it is in a suitable state for publication, hence the decision of Major Revisions. Both reviewers have provided a general review as well as notes on the PDF manuscript - please can you attend to all of their suggestions. In particular, I think it is important you consider and discuss the studies and taxa recommended by reviewer 1 and that you support the discussion of apomorphic and plesiomorphic characters with more evidence (reviewer 2). The introduction would benefit from a paragraph at the end setting out the aims of this study. Finally, the English language needs amending in many places - reviewer 1 has made extensive corrections, but there are still a number of other spelling and grammatical errors.

I look forward to reading the revised manuscript.

·

Basic reporting

Overall, the article was well written and clear. The introduction and background information are appropriate and generally thorough, the structure of the submission is appropriate, the figures are very well done, and all sources of data are clear and accessible. The descriptions of taxa and anatomy are clear and consistent, though a few errors (noted in the attached pdf) should be addressed.

However, there are a number (mostly) recent studies that should be considered and cited in the manuscript (Korth 1996, 1998, 2004, Rybczynski 2007, Korth and Samuels 2015, and Mörs et al. 2016). The most recent of those studies erects a new tribe Minocastorini, which includes Propalaeocastor, Oligotheriomys, Miotheriomys, Microtheriomys, and Minocastor. Similarly, Korth (1996) described Neatocastor, a genus of beavers he considered to be closely related to Agnotocastor, as such it should be considered here. Comparisons to all of those taxa should really be made as part of the results / discussion in this article. Since that Mörs et al. (2016) study just came out, it is understandable that it is not considered in this submission, but should be considered prior to publication.

Relevant citations follow:

Korth, W.W., 1996. A new genus of beaver (Mammalia: Castoridae: Rodentia) from the Arikareean (Oligocene) of Montana and its bearing on castorid phylogeny. Annals of Carnegie Museum, 65(2), pp.167-179.

Korth, W.W. 1998. A new beaver (Rodentia, Castoridae) from the Orellan (Oligocene) of North Dakota. Paludicola 1(4), pp.127-131.

Korth, W.W., 2004. Beavers (Rodentia, Castoridae) from the runningwater formation (early Miocene, early Hemingfordian) of western Nebraska. Annals of Carnegie Museum, 73, pp.1-11.

Rybczynski, N., 2007. Castorid phylogenetics: implications for the evolution of swimming and tree-exploitation in beavers. Journal of Mammalian Evolution, 14(1), pp.1-35.

Korth, W.W. and Samuels, J.X., 2015. New Rodent Material from the John Day Formation (Arikareean, Middle Oligocene to Early Miocene) of Oregon. Annals of Carnegie Museum, 83(1), pp.19-84.

Mörs, T., Tomida, Y. and Kalthoff, D.C., 2016. A new large beaver (Mammalia, Castoridae) from the early Miocene of Japan. Journal of Vertebrate Paleontology, 36(2), p.e1080720.

Experimental design

See comment above related to comparisons with other relevant taxa.

There are a few dental terms used in the text that are not consistent with what is labeled in Figure 2. The text repeatedly uses the term "entoloph", but Figure 2 shows the "endoloph" of the upper dentition. It also appears that several structures in Figure 2 are mislabeled. Specifically, the labels read "mesial hypolophulid", "medial hypolophulid", and "distal hypolophulid". In the text, these are all (correctly) referred to as "metalophule" instead.

Also, several terms used for cranial foramina are valid, but not consistent with the terms typically used by other authors (Wahlert 1972, 1977, Korth various studies, Rybczynski 2007). In particular, "caudal" palatine foramen and "major and minor" palatine foramen are used, but other authors typically refer to these as the dorsal palatine foramen and paired posterior palatine foramina instead. If the authors prefer to use the terms they have, they should at least note the alternative names used in other studies.

See the attached annotated manuscript PDF file for detailed comments.

Validity of the findings

Comparisons of Propalaeocastor irtyshensis to other members of the genus are well done. In the comparisons to other castorids, the recognized presence of a P3 (or dP3) is particularly relevant, and should be mentioned. As mentioned previously, since revision of Propalaeocastor is one of the stated goals of this study, the authors should be sure to include comparisons to all relevant castorids. To be fair, some of those comparisons are only necessary due to several very recent studies.

See the attached annotated manuscript PDF file for detailed comments.

Additional comments

Overall, I believe the study was well done. Propalaeocastor is an interesting taxon, and is particularly important for understanding the evolution and biogeography of beavers. Most of the issues I describe above and corrections noted in the annotated manuscript PDF file should be easily addressed. Comparisons to a broader range of taxa and recent phylogenetic placement of Propalaeocastor by other authors should help improve the utility of this study to readers.

Reviewer 2 ·

Basic reporting

If this article is fairly well structured, the writing style needs to be improved. A few suggestions and comments appear in the pdf file, but they are far from being exhaustive.

Experimental design

The scientific approach is rigorous and accurate regarding comparative anatomy and dental description. However, the author failed in expressing clearly their research question, probably due to the fact their study mainly relies on the sole description of additional material in order to define more precisely an old species of castorid.

Validity of the findings

There are only few original results given that the material described here belongs to a species already defined on dental characters. Nonetheless, the authors also described a few pieces of rostrum and drew interesting comparisons in the discussion between the described species, other castorids, and their extinct sister-group, the Eutypomyidae. Their discussion on plesiomorphic and apomorphic characters is a bit superficial and deserves to be based on more clearly stated and robust evidences. Phylogenetical analyses could help to solve this issue, as well as to precise the relationships between the oldest North American and Eurasian castorids and their putative events of migration. This approach will strengthen the interpretation made by the authors, and will also permit to more deeply discuss their systematic issues.

Additional comments

More generally, the authors should render their manuscript more accessible to readers in improving the writing style, in being more precise throughout the manuscript, notably in the abstract and the discussion, and in using a more appropriate method (e.g. phylogenetical analyses) to discuss the status of their characters.
Additional comments and suggestions are detailed in the pdf file.

Annotated reviews are not available for download in order to protect the identity of reviewers who chose to remain anonymous.

---

## Round 0.2 · Minor Revisions

· Academic Editor

Minor Revisions

Thank you for all the hard work you have put into revising this manuscript. I think it is much improved on the previous version. One of the reviewers of the original version has looked at the revised manuscript and provided comments and also made annotations on the manuscript itself. Please can you attend to these comments.

My main remaining concern with the manuscript is one of style. There seems to be a mixing of results and discussion. To me, your discussion starts where you currently have the subheading 'Comparisons'. You can then relabel the current 'Discussion' section as 'Phylogenetic analysis'. However, note that some of your phylogenetic analysis is very much reporting of results and so should be moved into the results section (as suggested by the reviewer).

Finally, I have made a number of changes to improve the language. These are in the second attached PDF.

I look forward to seeing your revised manuscript.

Reviewer 2 ·

Basic reporting

The authors put a lot of effort in performing a phylogenetical analysis, and in adding much more references. However, the new phylogenetical results should appear in the “results” part, not in the discussion. The writing style still needs to be strongly improved. Suggestions and comments appear in the pdf file.

Experimental design

If morphological descriptions are accurate, the phylogenetical analyses need more details concerning the definition of outgroup and character state ordering. Mention of consistency and retention indexes is also necessary. The authors should not discuss species attribution and synonymy in their descriptive part, but rather after presenting the results of their phylogenetical analyses.

Validity of the findings

The authors significantly improved their work in adding phylogenetical analyses to strengthen their discussion on plesiomorphic and apomorphic characters. However, they should more importantly use these results as well as characters defining important nodes for taxonomic discussion, comparison with previous studies on descriptions of extinct taxa, and definition of castorids (compared with Eutypomyidae). Moreover, final interpretations on the geographic origin of castorids, as well as radiation are very short and need supporting evidence. They should be more importantly discussed.
Additional comments and suggestions are detailed in the pdf file.
I hope that these last suggestions will help the authors to improve their manuscript.

Annotated reviews are not available for download in order to protect the identity of reviewers who chose to remain anonymous.

---

## Round 0.3 · accepted · Accept

· Academic Editor

Accept

Thank you for your revised manuscript. You have addressed all the comments from the reviewer and myself, so I now think this manuscript is ready for publication. Well done for all your hard work on this paper.